# Evaluating the predictive strength of the LACE index in identifying patients at high risk of hospital readmission following an inpatient episode: a retrospective cohort study

Sarah Damery,[1] Gill Combes[2]

[1]Institute of Applied Health Research, College of Medical and Dental Sciences, University of Birmingham, Edgbaston, UK
[2]CLAHRC West Midlands Research Lead for Chronic Diseases Theme, Institute of Applied Health Research, College of Medical and Dental Sciences, University of Birmingham, Edgbaston, UK

**Correspondence to**
Dr Sarah Damery; s.l.damery@bham.ac.uk

## ABSTRACT

**Objective** To assess how well the LACE index and its constituent elements predict 30-day hospital readmission, and to determine whether other combinations of clinical or sociodemographic variables may enhance prognostic capability.

**Design** Retrospective cohort study with split sample design for model validation.

**Setting** One large hospital Trust in the West Midlands.

**Participants** All alive-discharge adult inpatient episodes between 1 January 2013 and 31 December 2014.

**Data sources** Anonymised data for each inpatient episode were obtained from the hospital information system. These included age at index admission, gender, ethnicity, admission/discharge date, length of stay, treatment specialty, admission type and source, discharge destination, comorbidities, number of accident and emergency (A&E) visits in the 6 months before the index admission and whether a patient was readmitted within 30 days of index discharge.

**Outcome measures** Clinical and patient characteristics of readmission versus non-readmission episodes, proportion of readmission episodes at each LACE score, regression modelling of variables associated with readmission to assess the effectiveness of LACE and other variable combinations to predict 30-day readmission.

**Results** The training cohort included data on 91 922 patient episodes. Increasing LACE score and each of its individual components were independent predictors of readmission (area under the receiver operating characteristic curve (AUC) 0.773; 95% CI 0.768 to 0.779 for LACE; AUC 0.806; 95% CI 0.801 to 0.812 for the four LACE components). A LACE score of 11 was most effective at distinguishing between higher and lower risk patients. However, only 25% of readmission episodes occurred in the higher scoring group. A model combining A&E visits and hospital episodes per patient in the previous year was more effective at predicting readmission (AUC 0.815; 95% CI 0.810 to 0.819).

**Conclusions** Although LACE shows good discriminatory power in statistical terms, it may have little added value over and above clinical judgement in predicting a patient's risk of hospital readmission.

### Strengths and limitations of this study

► This study assessed the characteristics associated with 30-day hospital readmission in a large hospital Trust in the West Midlands.
► A split sample design allowed model development and statistical testing to be undertaken in one-half of the data set, and the results were validated in a representative sample of inpatient episodes from a directly comparable population.
► In focusing on a general medical population, the study evaluated the LACE index in a context similar to that in which it was originally developed.
► Readmission rates may have been underestimated as we were unable to identify cases where a patient may have been readmitted to another hospital.

## INTRODUCTION

In recent years, developing effective ways to reduce rates of patient readmission following an episode of acute care has become a key health policy focus in many developed economies.[1] In 2011/2012, the average 30-day readmission rate in England was 5.6%, with variation across acute Trusts of between 3% and 10%.[2] It is estimated that 30-day readmissions incur annual costs in excess of £2.5 billion for the National Health Service (NHS),[3] and in 2011 the Department of Health introduced a policy of non-payment to hospitals in England for emergency readmissions within 30 days of discharge following an elective admission. In 2012, this was extended to encompass both elective and emergency admissions. The policy applies to all clinical areas except those where it is considered inappropriate to withhold payment (eg, readmission in cancer or dialysis patients), and operates by establishing local readmission thresholds following clinical review of readmissions at a given Trust and determining the proportion that could be

considered avoidable.[4] In a financially straitened NHS, the prospect of incurring financial penalties for 30-day readmissions has created a strong incentive for hospital Trusts to reduce readmission rates.

The most common clinical reasons for readmission are infections and complications related to medical care or long-term conditions,[5] and it is thought that readmission rates can be reduced substantially if at-risk patients can be identified before discharge and offered supportive interventions as inpatients or after discharge. However, interventions are most likely to be effective if they are targeted towards patients at highest risk of future hospital use,[6] as being able to distinguish between patients who will not require readmission and those who are likely to be readmitted has implications for the cost-effectiveness of readmission avoidance interventions.[7–9] Identifying at-risk patients effectively relies on accurate case finding,[10] and a large number of predictive models have been used both within the NHS[3 11–13] and internationally,[14–16] to varying degrees of success.[17–19] Predictive models differ in the type and scope of data items they include and the time period over which they seek to predict readmission risk.[20] When choosing an appropriate model, a trade-off often needs to be made between complexity — the number of data items required — and practicality of application in clinical practice.[21]

One widely used predictive tool is the LACE index,[22] which was originally developed in Canada and uses routinely collected clinical and administrative data to generate a risk score of between 0 and 19 for individual patients, where higher scores indicate an increased risk of readmission. Scores are based on four features of an inpatient hospital episode: length of stay (LoS), admission type, comorbidities and the number of accident and emergency (A&E) visits made by a patient in the 6 months prior to their initial admission. Scores over a specific threshold can be used to 'flag' at-risk patients for whom interventions may be appropriate. Although widely used — largely due to its simplicity and the ease of LACE score calculation using data routinely collected by all hospital Trusts — the evidence base for LACE is uncertain. Some studies have found it to be an effective predictor of readmission,[23 24] whereas others have demonstrated poor prognostic ability, particularly when applied to specific patient subgroups.[25 26] The literature on case finding tools emphasises the importance of local validation before implementation, since each hospital has a patient casemix that reflects their surrounding population and may require a locally calibrated score threshold.[27] A modified version of the LACE index has been developed[27] that gives greater weight to patient comorbidities, which are considered a key driver of readmissions.[21] This study analysed data from a large hospital Trust in the West Midlands to assess how well the (modified) LACE index and each of its constituent elements predict 30-day readmission, and to determine whether a model based on other combinations of clinical or patient variables may enhance prognostic capability.

## METHODS

### Sampling

The study used a retrospective cohort design with a split sample to allow the findings to be externally validated. All alive-discharge adult inpatient episodes at a large hospital Trust in the West Midlands over a 2-year period (1 January 2013 to 31 December 2014) were included in the analysis. Data were obtained following a search of the Trust information system performed by the Trust information technology manager. Anonymised sociodemographic and clinical data were obtained for each inpatient episode (termed the 'index admission'). Sociodemographic data included patient age at index admission, gender and ethnic group. Clinical data relating to the index admission included date of admission and discharge, LoS, International Classification of Diseases (ICD)10 code, primary diagnosis, treatment specialty, Health Research Group code, admission type (emergency, elective, day case), admission source and discharge destination (eg, usual place of residence, other NHS institution). Data were also obtained on patient comorbidities, number of A&E visits in the 6 months before the index admission, whether the index episode was followed by readmission within 30 days of the index discharge date, and if so, the date of readmission and treatment specialty.

### Data analysis

A LACE score was calculated based on LoS (0–6 points), admission type (0–3 points), comorbidity (0–6 points) and previous A&E attendance (0–4 points), giving a total score of between 0 and 19 for each inpatient episode (table 1).

These scores differ from the original LACE index[22] in two ways. First, the original LACE index assigns up to seven points for an LoS lasting 14 or more days, whereas the modified LACE index gives up to six points for this parameter. Second, the comorbidity element of the original LACE index is scored up to a maximum of five, whereas the modified LACE index allows comorbidity scores of up to six points.

Any patient episodes with missing data were removed from the data set (n=4503 episodes; 2.4% of the total). Missing data items fell into three groups: (1) patients were discharged from their index hospital episode after 31 December 2014 (n=727), (2) no date of discharge from index hospital episode was available (n=661), and (3) patients died during their index admission (n=3115). After removal of records with missing data, the data set was split in half at random to create a cohort for model building (the 'training cohort') and a separate cohort for model validation (the 'test cohort'). As a split sample design was used to derive the two cohorts from the same original data set (ensuring patient and clinical profiles were directly comparable across the cohorts and minimising the likelihood of model overfitting), internal cross-validation within the training cohort was not performed during model development.

**Table 1** Components of the modified LACE index and values assigned for each

| Attribute | Value | Points |
|---|---|---|
| Length of stay | Less than 1 day | 0 |
| | 1 day | 1 |
| | 2 days | 2 |
| | 3 days | 3 |
| | 4–6 days | 4 |
| | 7–13 days | 5 |
| | 14 or more days | 6 |
| Acute admission | Inpatient | 3 |
| | Observation | 0 |
| Comorbidity (scores cumulative to a maximum of 6) | No history | 0 |
| | Diabetes without complications, cerebrovascular disease, history of myocardial infarction, peripheral vascular disease, peptic ulcer disease | 1 |
| | Mild liver disease, diabetes with end organ damage, congestive heart failure, chronic obstructive pulmonary disease, cancer, leukaemia, lymphoma, any tumour, cancer, moderate to severe renal disease | 2 |
| | Dementia or connective tissue disease | 3 |
| | Moderate/severe liver disease or HIV infection | 4 |
| | Metastatic cancer | 6 |
| Accident and emergency visits during the previous 6 months | 0 visit | 0 |
| | 1 visit | 1 |
| | 2 visits | 2 |
| | 3 visits | 3 |
| | 4 or more visits | 4 |

Normality testing indicated that the continuous data were not normally distributed. As a result, all continuous variables were summarised using medians and IQRs, and univariate comparisons of these variables across the readmitted/non-readmitted groups used the non-parametric Mann-Whitney U test. $X^2$ tests were used to compare the characteristics of readmitted versus non-readmitted patients for variables with categorical data. Univariate ORs and their 95% CIs were calculated for sociodemographic and clinical variables and for each component of the LACE index to test the association between variable subgroups and readmission. Finally, binary logistic regression modelling using the enter method was used to test the strength of different combinations of variables in predicting the likelihood of 30-day readmission. Model strength was described using OR, and the area under the receiver operating characteristic curve (AUC) was described using the *c*-statistic. The findings for each model were then validated using the patient episodes in the test cohort. All statistical analyses were undertaken using SPSS V.21.

## RESULTS
### Sample characteristics
The full data set included 183 843 patient episodes (103 493 individual patients). After splitting the data set to create the training and test cohorts, subsequent analyses were performed on the training cohort prior to validation, which contained data on 91 922 separate admission episodes (representing 51 747 individual patients) (table 2).

The median patient age in the training cohort was 55 (IQR: 37–72), and male patients accounted for 42.4% of hospital episodes (n=39 001). The median LoS of the index admission was 0 days (IQR: 0–2). Fifty-one per cent of the episodes followed emergency admission (n=46 922). The majority of patients had a comorbidity score of 0 (80.8%), and the median number of A&E visits in the 6 months prior to the index episode was 1 (IQR: 0–1).

### Characteristics of readmitted versus non-readmitted patients
A total of 7107 inpatient episodes were followed by a readmission within 30 days (7.7% readmission rate, 4541 individual patients). One thousand and two hundred eighteen (2.4%) patients accounted for 53.1% of all readmission episodes. A comparison of the characteristics of episodes that resulted in readmission versus those that did not showed statistically significant differences for all variables. Readmitted patients were significantly more likely to be older than those who were not readmitted (median age 64 vs 55), men had significantly higher readmission rates than women (9.1% vs 6.7%), and emergency admissions were significantly

**Table 2** Characteristics of readmission versus non-readmission episodes

| Variable | Grouping | Total episodes (%) | Readmitted (%) | Not readmitted (%) | Comparison* |
|---|---|---|---|---|---|
| Patient age | Median, range (IQR) | 55.0, 18–106 (37–72) | 64.0, 18–105 (44–78) | 55.0, 18–106 (37–71) | p<0.0001 |
| Gender | Male | 39 001 (42.4) | 3545 (9.1) | 35 456 (90.9) | $X^2$=175.1; p<0.0001 |
| | Female | 52 921 (57.6) | 3562 (6.7) | 49 359 (93.3) | |
| Index length of stay (days) | Median, range (IQR) | 0.0, 0–301 (0.0–2.0) | 1.0, 0–223 (0.0–5.0) | 0.0, 0–301 (0.0–1.0) | p<0.0001 |
| Admission type | Emergency | 46 922 (51.0) | 6005 (12.8) | 40 917 (87.2) | $X^2$=3573.4; p<0.0001 |
| | Elective | 7243 (7.9) | 410 (5.7) | 6833 (94.3) | |
| | Day case | 37 757 (41.1) | 692 (1.8) | 37 065 (98.2) | |
| Comorbidity score† | 0 | 74 274 (80.8) | 5083 (6.8) | 69 191 (93.2) | $X^2$=1126.9; p<0.0001 |
| | 1 | 14 984 (16.3) | 1820 (12.1) | 13 164 (87.9) | |
| | 2 | 2147 (2.3) | 29 (1.4) | 2118 (98.6) | |
| | 3 | 514 (0.6) | 172 (33.5) | 342 (66.5) | |
| | 4 | 3 (0.0) | 3 (100.0) | 0 (0.0) | |
| | 6 | 0 (0.0 | 0 (0.0) | 0 (0.0) | |
| Accident and emergency visits in the previous 6 months | Median, range (IQR) | 1.0, 1–121 (1.0–2.0) | 2.0, 1–107 (1.0–4.0) | 1.0, 1–121 (1.0–2.0) | p<0.0001 |

*Continuous variables were compared using the Mann-Whitney U test and categorical variables were compared using the $X^2$ test.
†Comorbidity score does not relate to the number of comorbidities; scores are assigned based on severity of comorbidities.

more likely to result in readmission than elective or day case admissions (12.8% vs 5.7% and 1.8%, respectively). The median LoS in readmitted patients was 1 day (IQR: 0–5), the median A&E visits in the previous 6 months was 2 (IQR: 1–4), and higher comorbidity scores were significantly associated with readmission, with 33.5% of patients with a comorbidity score of 3 being readmitted within 30 days.

### Readmission episodes and LACE score
The median LACE score in the training cohort was 5 (range 3–17, IQR: 3–7) (table 3). The Mann-Whitney U test showed that the median score for readmission episodes was significantly higher than the median score for episodes that did not result in readmission (8.0 with IQR 6–11 vs 5.0 with IQR 3–7; p<0.0001). Readmission rates more than doubled between a LACE score of 10 and 11, suggesting that 11 may be the optimum threshold for distinguishing between patients at a lower or higher risk of readmission. However, the proportion of total readmissions represented by LACE scores 11 and above was only 25.3% of the total (1795/7107); thus, nearly three-quarters of readmissions occurred in patients scoring lower than the cut-off point.

### Univariate logistic regression
Univariate binary logistic regression assessed the association between individual variables and the likelihood of readmission (table 4). All variables were statistically significant to the p<0.0001 level. For each unit increase in patient age, the likelihood of readmission rose by 1.7%. Women were significantly less likely to

be readmitted than men, despite constituting a larger proportion of index admissions (OR 0.72; 95% CI 0.69 to 0.76). Increasing LACE score was significantly

**Table 3** Proportion of readmission episodes at each LACE score

| LACE score* | Total episodes (%) | Readmitted (%) | Not readmitted (%) |
|---|---|---|---|
| 3 | 26 478 (28.8) | 302 (1.1) | 26 176 (98.9) |
| 4 | 13 798 (15.0) | 561 (4.1) | 13 237 (95.9) |
| 5 | 14 152 (15.4) | 676 (4.8) | 13 476 (95.2) |
| 6 | 10 656 (11.6) | 753 (7.1) | 9903 (92.9) |
| 7 | 8637 (9.4) | 1045 (12.1) | 7592 (87.9) |
| 8 | 5800 (6.3) | 913 (15.7) | 4887 (84.3) |
| 9 | 4136 (4.5) | 551 (13.3) | 3585 (86.7) |
| 10 | 3242 (3.5) | 511 (15.8) | 2731 (84.2) |
| 11 | 2379 (2.6) | 815 (34.3) | 1564 (65.7) |
| 12 | 1570 (1.7) | 633 (40.3) | 937 (59.7) |
| 13 | 751 (0.8) | 231 (30.8) | 520 (69.2) |
| 14 | 217 (0.2) | 81 (37.3) | 136 (62.7) |
| 15 | 104 (0.1) | 35 (33.7) | 69 (66.3) |
| 16 | 1 (0.0) | 0 (0.0) | 1 (100.0) |
| 17 | 1 (0.0) | 0 (0.0) | 1 (100.0) |

*Percentages for readmitted and not readmitted are calculated according to variable grouping, for example, % of episodes scoring 3 on the LACE index which resulted in readmission versus those that did not.

**Table 4** Univariate logistic regression of variables potentially associated with readmission

| Variable | Grouping | p Value | OR (95% CI) |
|---|---|---|---|
| Patient age | Continuous | <0.0001 | 1.03 (1.02 to 1.04) |
| Patient gender | Male | Reference | Reference |
| | Female | <0.0001 | 0.72 (0.69 to 0.76) |
| Length of stay | Continuous | <0.0001 | 1.04 (1.03 to 1.04) |
| Admission type | Day case | Reference | Reference |
| | Elective | <0.0001 | 3.21 (2.84 to 3.64) |
| | Emergency | <0.0001 | 7.87 (7.26 to 8.52) |
| Comorbidity score | Continuous | <0.0001 | 1.28 (1.25 to 1.31) |
| Accident and emergency visits in the previous 6 months | Continuous | <0.0001 | 1.39 (1.38 to 1.41) |
| LACE score | Continuous | <0.0001 | 1.42 (1.41 to 1.43) |
| Episodes per patient in the previous year | Continuous | <0.0001 | 1.06 (1.05 to 1.06) |

associated with 30-day readmission, with each point increase in score associated with a 42% increase in the likelihood of readmission. The variable with the strongest association with readmission was emergency admission — index episodes that were a result of emergency admission were nearly eight times more likely to be followed by readmission than those in the reference group of day case surgery (OR 7.87; 95% CI 7.26 to 8.52).

## Multivariate logistic regression

Four multivariate logistic regression models were constructed to assess different potential predictors of 30-day readmission (table 5). The first included LACE index score only. LACE score was highly significant as a predictor of readmission (p<0.0001), with an AUC of 0.773 (95% CI 0.768 to 0.779) and $R^2$ of 0.180. This is higher than the $c$-statistic for LACE score as a predictor

**Table 5** Binary logistic regression models assessing the probability of readmission

| Variable | p Value | OR (95% CI) | AUC (95% CI); $R^2$ |
|---|---|---|---|
| **Model 1: LACE score only** | | | |
| LACE score | <0.0001 | 1.42 (1.41 to 1.43) | AUC=0.773 (0.768 to 0.779); $R^2$=0.180 |
| **Model 2: all variables from univariate analysis** | | | |
| Age | <0.0001 | 1.01 (1.01 to 1.02) | AUC=0.820 (0.815 to 0.825); $R^2$=0.240 |
| Gender (female) | <0.0001 | 0.90 (0.85 to 0.95) | |
| Length of stay | <0.0001 | 0.98 (0.98 to 0.99) | |
| Admission type (emergency) | <0.0001 | 4.18 (3.77 to 4.64) | |
| Comorbidity score | <0.0001 | 0.96 (0.94 to 0.98) | |
| A&E visits in the previous 6 months | <0.0001 | 1.12 (1.11 to 1.13) | |
| LACE score | <0.0001 | 1.23 (1.21 to 1.25) | |
| Episodes per patient | <0.0001 | 1.07 (1.06 to 1.07) | |
| **Model 3: four components of the LACE index** | | | |
| Length of stay | <0.0001 | 1.02 (1.01 to 1.03) | AUC=0.806 (0.801 to 0.812); $R^2$=0.193 |
| Admission type (emergency) | <0.0001 | 4.79 (4.40 to 5.22) | |
| Comorbidity score | <0.0001 | 1.30 (1.26 to 1.33) | |
| A&E visits in the previous 6 months | <0.0001 | 1.28 (1.27 to 1.29) | |
| **Model 4: reduced complexity model** | | | |
| A&E visits in the previous 6 months | <0.0001 | 1.36 (1.35 to 1.38) | AUC=0.815 (0.810 to 0.819); $R^2$=0.130 |
| Episodes per patient in the previous year | <0.0001 | 1.03 (1.02 to 1.04) | |

A&E, accident and emergency; AUC, area under the receiver operating characteristic curve.

of readmission found by van Walraven *et al*, who developed the LACE index. However, the other three models all had a higher *c*-statistic: model 2, which included all eight variables from the univariate analysis had a *c*-statistic of 0.820 (95% CI 0.815 to 0.825) and $R^2$ of 0.240. In this model, while all included variables were significant at the p<0.0001 level, the direction of effect for some variables differed from the univariate testing. Both increasing LoS and higher comorbidity score were associated with a *lower* likelihood of readmission in this model when the effects of other variables were controlled for. This is likely to be due to an association between patient age and/or gender with comorbidities and LoS. Model 3 included the four components of the LACE index and was a better predictor of readmission than LACE score alone, with a *c*-statistic of 0.806 (0.801–0.812). The fourth model, which included only A&E visits and number of episodes per patient as variables, was a better model for predicting readmission than models 1 and 3 and had only a marginally lower *c*-statistic than model 2, which was the most complex in terms of the number of variables included (AUC=0.815; 95% CI 0.810 to 0.819).

### Model validation

The models developed using the training cohort were tested for validity in the test cohort. The test cohort did not differ from the training cohort in any of the socio-demographic of clinical variables assessed. It included 91921 episodes of care, of which 7008 (7.6%) were followed by a readmission within 30 days. The *c*-statistic of the logistic regression models developed from the training cohort were 0.767 (0.761–0.772) for model 1, 0.814 (0.809–0.189) for model 2, 0.800 (0.794–0.805) for model 3 and 0.812 (0.807–0.817) for model 4.

### DISCUSSION

The primary aim of this study was to assess how well the modified LACE index was able to predict 30-day readmission in a cohort of patients admitted to a large secondary care Trust over a 2-year period. Increasing LACE score and the four individual components comprising the LACE index were all independent predictors of readmission. The proportion of admissions episodes resulting in readmission increased substantially at a LACE score cut-off of 11, which would suggest that this is an appropriate threshold to use when deciding whether to provide enhanced inpatient and/or postdischarge care to prevent unplanned readmission. However, although a large proportion of admissions episodes that scored 11+ on the LACE index were followed by a readmission within 30 days, this corresponded to comparatively few absolute numbers of patients. Only 25% of all readmissions occurred in the higher scoring group, while the remaining 75% occurred following episodes of care that scored <11 on the index. This differs from other studies that have assessed the effectiveness of different risk thresholds for LACE, which typically saw a higher

proportion of all readmissions occurring in the patient group that scored above the chosen threshold.[23 28] While implementing a lower LACE score threshold would improve the likelihood of identifying at-risk patients, a large number of these patients would not go on to be readmitted. In a health service facing substantial resource constraints, the LACE tool is unlikely to have the sensitivity and specificity that would make it a useful addition to clinical practice.

A number of studies have assessed the performance of the LACE index in predicting unplanned readmissions, but these have typically been conducted in small patient populations,[29 30] or in specific patient groups such as cardiovascular disease,[25 29 31] chronic obstructive pulmonary disease[30] or older people.[26] The patient cohort included in this study was large, and the analysis had good statistical power to detect differences between groups. In focusing on a general medical population, our study evaluated the LACE index in a context similar to that in which it was originally developed in terms of patient characteristics and incidence of comorbidity.[22 27] The split sample design allowed model development and statistical testing to be carried out in one-half of the data set, and the results were validated in a representative sample of inpatient episodes from a directly comparable population. Readmission rates may have been underestimated in the hospital data used for this study, as we were unable to identify instances where a discharged patient may have subsequently been readmitted to another hospital. Patient deaths were not recorded in the data (unless a patient died during index admission), so we were unable to consider the impact of patient mortality on our findings.

Multiple factors typically contribute to readmission rates, and there are limits on the extent to which unplanned readmissions can be avoided.[32 33] High readmission rates are often thought to indicate suboptimal patient management, but they are most likely to be driven by difficulties in managing patient transitions to other health and social care settings, a lack of community resources for patient follow-up, or influenced by the patient's home environment.[34] A retrospective analysis of 82 million routinely collected hospital records in England between 2004 and 2010 found that only 30% of unplanned readmissions were deemed avoidable.[35] Therefore, lowering readmission rates for patients with chronic or relapsing conditions, or patients readmitted with a different diagnosis from their index admission, poses a significant challenge. Conversely, avoiding readmissions in patients presenting with a recurrence or continuation of the issue that led to their initial hospitalisation, or for those who are readmitted with an avoidable complication related to their index admission, should be a priority for hospital Trusts, which is a key reason that case finding tools are increasingly being tested in the hospital setting. Although a number of increasingly complex tools have been developed in recent years, such as PARR-30,[36] LACE+[37] and HOSPITAL,[38] the intuitive

appeal of LACE lies in its simplicity and use of routinely collected hospital data.

This study suggests that despite a number of socio-demographic and clinical variables being strongly associated with hospital readmission in statistical terms, the added value of the LACE tool over and above clinical judgement remains equivocal. However, the fact that small gains in model accuracy and discriminatory power can be made by testing different combinations of potential predictor variables derived from routinely collected hospital administrative data may indicate that the accuracy of case finding could be improved through the addition of locally relevant clinical or sociodemographic factors.[39 40] In this study, the predictive model with the least discriminatory power was based on LACE score alone. Model 2, which included eight predictor variables, was only marginally better at predicting readmission than model 4, which included only two variables: A&E visits and the number of admissions per patient in the previous 12 months. This would suggest — in a cohort of general medical admissions — that a simpler model could outperform the more complex LACE tool in accurately identifying patients at risk of readmission. Our analysis showed that 2.4% of patients in the cohort accounted for 53.1% of all readmission episodes. Being able to identify the small group of patients who use a disproportionate amount of healthcare resources is the first step towards developing solutions to prevent repeat hospitalisations in this population.[41] Future research should focus on the development of locally tailored screening tools to identify these patients.

## CONCLUSION

Although LACE shows good discriminatory power in statistical terms, it may have little added value over and above clinical judgement in predicting a patient's risk of hospital readmission. Nevertheless, if used as a screening tool alongside clinical judgement, a locally tailored risk score based on specific clinical or sociodemographic variables relevant to the inpatient population admitted to a particular hospital Trust may increase case finding accuracy. This could allow clinicians to effectively discriminate between patients who are likely to have an unplanned admission within 30 days of discharge and those who will not.

**Acknowledgements** We would like to thank Martin Chadderton for providing the data from the hospital information system, and Roger Steadman for clinical guidance on the study.

**Contributors** SD and GC designed the study. SD wrote the study protocol. SD undertook data analysis, with input from GC as needed. SD drafted and revised the paper and is the guarantor for the work. GC critically revised the paper for intellectual content. Both authors gave final approval of the manuscript and are accountable for all aspects of the accuracy and integrity of the work.

**Funding** This research was funded by the National Institute of Health Research (NIHR) Collaboration for Leadership in Applied Health Research and Care West Midlands (CLAHRCWM).

**Competing interests** None declared.

**Patient consent** Detail has been removed from this case description/these case descriptions to ensure anonymity. The editors and reviewers have seen the detailed information available and are satisfied that the information backs up the case the authors are making.

**Ethics approval** Ethical approval was obtained from the University of Birmingham Research Ethics Committee (Ref: ERN_14-0914). Research governance approval was obtained from the R&D office of Sandwell and West Birmingham Hospitals NHS Trust (Ref: 14MISC40).

**Provenance and peer review** Not commissioned; externally peer reviewed.

**Data sharing statement** No additional data are available.

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
