## [Reviewer comments · BMJ Open]

ARTICLE DETAILS

TITLE (PROVISIONAL)	Evaluating the predictive strength of the LACE index in identifying patients at high risk of hospital readmission following an inpatient episode: retrospective cohort study
AUTHORS	Damery, Sarah; Combes, Gill

VERSION 1 - REVIEW

REVIEWER	Paolo Eusebi University Hospital of Perugia, Italy
REVIEW RETURNED	04-Apr-2017

GENERAL COMMENTS	The authors removed "any patient episodes with missing data". I wonder if missing data could be treated somewhat and if the authors could report more information on the missing values patterns. Instead of using the term "derivation cohort for statistical testing" replace with "cohort for model building" or "training cohort". Please report if internal cross-validation was also performed, if not justify.
--

REVIEWER	Robert Robinson SIU School of Medicine USA
REVIEW RETURNED	18-Apr-2017

GENERAL COMMENTS	Well designed and implemented study.
--------------------------------------

REVIEWER	Christo El Morr York University CANADA
REVIEW RETURNED	23-Apr-2017

GENERAL COMMENTS	Evaluating the predictive strength of the LACE index Introduction 1 - "alive-discharge adult": how did the authors get the alive patients? Did they have any access to outsider patient? What is the source of the data? 2- The authors make good overview of the available models to predict readmissions; however, their choice of LACE is based on its "widely used predictive". The reader would like to know if there are
--

other reasons for choosing LACE; for instance, authors mention that some of the models are complex; is it that LACE less complex than other models? This should be stated clearly; especially that LACE was developed in Canada while other models were developed locally (in the UK).

3- Authors make excellent point when they declare that a model “require a locally-calibrated score threshold.”.

Methods

TABLE 1

1- For L value “14 or more days” the number of points show 6; in LACE the number of points in 7; if this modified tool changed L values please indicate it clearly.

2- The introduction let the reader think that the modified LACE index that the authors are using differ in the weight given to comorbidities; however, looking at Table 1 the comorbidities are assigned the same number of Points as in the original LACE index. Please clarify.

3- “Continuous variables were compared using the Mann-Whitney U-test”; please explain why? why not the parametric t-test? did you test for normalization and found the distribution not-normal? If so; what factor led to your decision?

Results

Readmission episodes and LACE score

1- “The median score for readmission episodes was significantly higher than the median score for episodes that did not result in readmission (8.0 with IQR 6 to 11 vs. 5.0 with IQR 3 to 7; $p < 0.0001$ ”). What is the test that was used? Mann Whitney between scores of admitted and readmitted? Not clear; please clarify?

Discussion

1- in the discussion the authors mention “which typically saw a higher proportion of all readmissions occurring in the patient group that scored above the chosen threshold” and the threshold in question is 11; to support their point they refer to Gruneir study “Unplanned readmissions after hospital discharge among patients identified as being at high risk for readmission using a validated predictive algorithm” (reference 23); however, Grunier study had 10 as the threshold. Which is in fact supports what the authors found: basically that they might need a lower threshold ; if the authors take 10 as the threshold they would add 15.8% of readmissions, which raises the total to nearly 41% of readmission (25%+15.8%)

General

1- In the results you have mentioned that “1218 patients (2.4%) accounted for 53.1%”. what do you make out of this observation? Is there any identification of these patients at this stage? This is not essential as I understand that you have mentioned that you will be doing more work to identify these patients’ characteristics (“Future research should focus on the development of locally-tailored screening tools to identify these patients.”).

2- three quarters of readmissions occurred in patients scoring lower than the cut-off point (score 11). what do you make out of this observation? Should we reduce the LACE score cut-off to lower than 11? Efore making final conclusions I suggest the authors measure

	the odds-ratio for LACE cut off of 11 and 10 and alternatively 9 and see if the discriminatory power of the LACE threshold will be better at lower values; something similar has been performed by El Morr C, Ginsburg L, Nam S, Woollard S Assessing the Performance of a Modified LACE Index (LACE-rt) to Predict Unplanned Readmission After Discharge in a Community Teaching Hospital Interact J Med Res 2017;6(1):e2 https://www.i-jmr.org/2017/1/e2/ see paragraph (“Predictive Power of LACE-rt at the Hospital: High-Risk Versus Low-Risk Patients”) 3- the original LACE score was developed to predict death and readmission; have the authors looked at death? Did they subtract the people who died from their data set (the introduction let the reader think so: “All alive-discharge adult inpatient episodes at a large hospital”)? If it was the case then this might have skewed the results (underestimating the admission rates). This must be mentioned in the limitations. It adds to your conclusion that in your study “Readmission rates may have been underestimated in the hospital data used for this study, as we were unable to identify instances where a discharged patient may have subsequently been readmitted to another hospital”. 4- how would you explain the fact that “Both increasing length of stay and higher comorbidity score were associated with a lower likelihood of readmission in this model when the effects of other variables was controlled for.”?
--	---

VERSION 1 – AUTHOR RESPONSE

Reviewer: 1

Comments to the Authors:

1. The authors removed “any patient episodes with missing data”. I wonder if missing data could be treated somewhat and if the authors could report more information on the missing values patterns.

More information has been added on the missing values that resulted in patient episodes being removed from the dataset (page 6 – penultimate paragraph of methods section). Reasons for removal were: patient was discharged from their index episode after 31/12/2014 (i.e. outside of the study period), no discharge date from the index episode was available (i.e. whether a patient was readmitted within 30 days of index discharge could not be determined), and patient died during their index admission.

2. Instead of using the term “derivation cohort for statistical testing” replace with “cohort for model building” or “training cohort”.

Thank you for this suggestion – the text has been changed to ‘training cohort’ throughout (with the other cohort referred to as the ‘test cohort’).

3. Please report if internal cross-validation was also performed, if not justify.

It is our understanding that internal cross-validation is most useful when a dataset is too small to allow for a split sample design that creates similar training and testing cohorts (thus model development and testing is carried out on the ‘same’ data). Our model(s) had a small number of parameters, and

the dataset was large enough to allow being randomly split into two equal sized training and test cohorts. Thus, both cohorts came from the same 'parent' dataset and the profile of patients/patient characteristics did not differ between the two cohorts. The similarity in AUC values between models applied in the training and test cohorts suggested that the data were robust and not subject to bias from over fitting. Bearing all this in mind, we considered internal cross-validation when developing the models to be unnecessary, and internal cross-validation was therefore not performed. This is described in the final paragraph of the methods section (page 6).

Reviewer: 2

Comments to the Authors:

No comments received.

Reviewer: 3

Comments to the Authors:

1. "alive-discharge adult": how did the authors get the alive patients? Did they have any access to outsider patient? What is the source of the data?

The dataset was derived from the hospital information system at the participating NHS Trust. This is described at the start of the methods section. The existing wording has been changed slightly to clarify the source of the data. We are not clear what the reviewer means by 'access to outsider patient' so cannot address this point. All patients were admitted to, and discharged from a single NHS Trust. We believe this is clearly stated in the manuscript.

2. The authors make good overview of the available models to predict readmissions; however, their choice of LACE is based on its "widely used predictive". The reader would like to know if there are other reasons for choosing LACE; for instance, authors mention that some of the models are complex; is it that LACE less complex than other models? This should be stated clearly; especially that LACE was developed in Canada while other models were developed locally (in the UK).

The fact that LACE was originally developed in Canada is now mentioned in the introduction. The primary reason for choosing LACE is that it can be calculated easily from routinely collected hospital data and thus does not require any additional data collection to be undertaken before it can be used. We already mention this in the discussion, but have added the point to the introduction that the simplicity of the tool and ease of scoring are the main sources of its appeal (extra sentence added to the final paragraph of the introduction – page 4).

3. Authors make excellent point when they declare that a model "require a locally-calibrated score threshold".

Thank you for recognising this.

4. Table 1: For L value "14 or more days" the number of points show 6; in LACE the number of points is 7; if this modified tool changed L values please indicate it clearly.

The reviewer is correct – the original LACE tool as developed by Carl van Walraven et al scored length of stay up to 7 points, whereas the modified LACE tool used in our study only gives a maximum of 6 points for length of stay. Differences between the original LACE index and the modified version used in our study are now described at the top of page 6.

5. The introduction let the reader think that the modified LACE index that the authors are using differ

in the weight given to comorbidities; however looking at Table 1 the comorbidities are assigned the same number of points as in the original LACE index. Please clarify.

We think the reviewer is incorrect here – although the original LACE index does include a score up to 6 for comorbidities, only a maximum of 5 points can be taken forward for the comorbidities element of the score. In contrast, the modified LACE index allows the full maximum of 6 points to be taken forward for the comorbidities part of the score – thus the modified LACE index does indeed give more weight to comorbidities in the overall score. No changes have been made to the text following this comment.

6. “Continuous variables were compared using the Mann-Whitney U-Test”. Please explain why? Why not the parametric t-test? Did you test for normalization and found the distribution not-normal? If so; what factor led to your decision?

The reviewer is correct in his assumption – we tested the data for normality and found that data were not normally distributed, therefore the parametric t-test was not appropriate. As the Mann Whitney U-test is the non-parametric equivalent of the t-test, this explains our choice of this statistical test. We have added detail to the final paragraph of the methods section (page 6) to explain that we carried out normality testing.

7. “The median score for readmission episodes was significantly higher than the median score for episodes that did not result in readmission (8.0 with IQR 6 to 11 vs. 5.0 with IQR 3 to 7; $p < 0.0001$)”. What is the test that was used? Mann Whitney between scores of admitted and readmitted? Not clear; please clarify.

The analysis that the reviewer refers to was undertaken using the Mann Whitney U-test. This has been clarified in the text (final paragraph of page 7).

8. In the discussion the authors mention “which typically saw a higher proportion of all readmissions occurring in the patient group that scored above the chosen threshold” and the threshold in question is 11; to support their point they refer to Gruneir study “Unplanned readmissions after hospital discharge among patients identified as being at high risk for readmission using a validated predictive algorithm” (reference 23); however, Gruneir study had 10 as the threshold. Which in fact supports what the authors found: basically that they might need a lower threshold; if the authors take 10 as the threshold they would add 15.8% of readmissions, which raises the total to nearly 41% of readmission (25% + 15.8%).

Apologies for confusion here. We were not stating that Gruneir used 11 as their threshold, but we were making the point that Gruneir is an example of a study where – whatever the chosen threshold in the study – a high proportion of readmissions occurred in patients scoring above the threshold compared to readmissions in patients scoring below the threshold i.e. that a specific cut-off of LACE was able to discriminate well between readmitted and not readmitted patients. Our data do not show the same distinction between readmitted vs. not readmitted patients at different thresholds.

The reviewer is correct that reducing our cut-off to 10 would increase the proportion of readmissions detected. However, this would not increase the proportion detected to 41% of readmissions as the reviewer suggests: proportions in Table 3 are those at each LACE score. So, reducing the threshold to 10 would identify another 511 readmissions episodes (7.1% of total readmissions), taking the total readmissions correctly identified to 32.4% (25% + 7.1%). Unfortunately, doing so would also increase substantially the proportion of non-readmissions detected, by adding another 2730 patient episodes that did not result in readmission (a total of 7% of non-readmissions at cut-off score of 10 compared to 3.8% of non-readmissions at a cut-off score of 11). This suggests that 11 is the most statistically

discriminatory cut-off in our dataset, but does not change our overall conclusion that LACE performs poorly in our patient cohort as an unacceptably high proportion of total readmissions occur in patients with LACE scores below 11. We have not made any changes to the manuscript in the light of this comment.

9. In the results you have mentioned that “1218 patients (2.4%) accounted for 53.1%”. What do you make out of this observation? Is there any identification of these patients at this stage? This is not essential as I understand that you have mentioned that you will be doing more work to identify these patients’ characteristics (“Future research should focus on the development of locally-tailored screening tools to identify these patients”).

Thank you for this point. We have chosen not to address it in this paper, because – as the reviewer points out – this is something that we plan to cover in future work i.e. the purpose of the current paper is to look at how well LACE performs on our dataset. Future work will look at whether there are other characteristics that could be included in a locally-tailored model to improve prediction of readmissions. No changes have been made to the manuscript as a result of this comment.

10. Three quarters of readmissions occurred in patients scoring lower than the cut-off point (score 11). What do you make out of this observation? Should we reduce the LACE score cut-off to lower than 11? Before making final conclusions I suggest the authors measure the odds-ratio for LACE cut-off of 11 and 10 and alternatively 9 and see if the discriminatory power of the LACE threshold will be better at lower values; something similar has been performed by El Morr et al 2017 (reference supplied).

Please see our response to point 8 above. Our analysis suggests that reducing the cut-off score for LACE to 10, or even 9 would not improve the discriminatory power of the tool to distinguish between patients who were readmitted and those who were not. We did carry out the analysis suggested by the reviewer when we were developing our manuscript, but chose not to include it as it did not alter our conclusions, and determining a cut-off for LACE was not a main outcome in the paper - the primary aims being a) to assess how well the LACE index and each of its constituent elements predicts 30-day readmission and b) to determine whether a model based on other combinations of variables may enhance prognostic capability. For the reviewer’s information, the OR for a cut-off of 11 was 8.54; the OR for a cut-off of 10 was 6.36 and the OR for a cut-off of 9 was 5.30. Although interesting, we do not feel that the argument in the paper would be enhanced by including odds ratios for different cut-offs of LACE so we have not made any changes to the manuscript in the light of this comment.

11. The original LACE score was developed to predict death and readmission; have the authors looked at death? Did they subtract people who died from their data set (the introduction lets the reader think so: “All alive-discharge adult inpatient episodes at a large hospital”)? If it was the case then this might have skewed the result (underestimating the admission rates). This must be mentioned in the limitations. It adds to your conclusion that in your study “Readmission rates may have been underestimated in the hospital data used for this study, as we were unable to identify instances where a discharged patient may have subsequently been readmitted to another hospital”.

Thank you for this point. Unfortunately, we did not have information on deaths in our dataset where death happened after discharge from the index inpatient episode. As mentioned in our response to reviewer 1 (point 1), our only mortality information related to cases where a patient died before being discharged from their index episode. These patients were removed from the dataset as they could not be readmitted if they had died before index discharge. We do not believe that removing these patients from the dataset is a limitation of the study. However, the reviewer is right to suggest that not capturing whether a patient died after their index discharge or after a readmissions episode is a

limitation of the study, and we have acknowledged this in our discussion (top of page 11).

12. How would you explain the fact that “Both increasing length of stay and higher comorbidity score were associated with a lower likelihood of readmission in this model when the effects of other variables was controlled for”?

Any explanation would be speculative, so we are reluctant to make any assumptions that cannot be proved by the data. However, the finding is likely to be due to the effect of controlling for age and gender in the model that the reviewer refers to, since both the incidence of comorbidities and length of stay are associated with patient age and gender. We have added a tentative explanation to the sentence that the reviewer mentions (page 9).

VERSION 2 – REVIEW

REVIEWER	Christo El Morr York University, Canada
REVIEW RETURNED	05-May-2017
GENERAL COMMENTS	The edits were satisfactory. Just note that the hyper link in references 2 and 27 are broken and need updating.